# Visual-Inertial Cross Fusion: A Fast and Accurate State Estimation Framework for Micro Flapping Wing Rotors

**Xin Dong** [1], **Ziyu Wang** [1], **Fangyuan Liu** [1], **Song Li** [1], **Fan Fei** [2], **Daochun Li** [1] and **Zhan Tu** [1,3,*]

1 School of Aeronautic Science and Engineering, Beihang University, Beijing 100191, China; xindong324@buaa.edu.cn (X.D.); wangziyu_95@163.com (Z.W.); fang_yuan_liu@buaa.edu.cn (F.L.); sy2005123@buaa.edu.cn (S.L.); lidc@buaa.edu.cn (D.L.)
2 School of Mechanical Engineering, Purdue University, West Lafayette, IN 47906, USA; feif@purdue.edu
3 Institute of Unmanned System, Beihang University, Beijing 100191, China
* Correspondence: zhantu@buaa.edu.cn

**Abstract:** Real-time and drift-free state estimation is essential for the flight control of Micro Aerial Vehicles (MAVs). Due to the vibration caused by the particular flapping motion and the stringent constraints of scale, weight, and power, state estimation divergence actually becomes an open challenge for flapping wing platforms' longterm stable flight. Unlike conventional MAVs, the direct adoption of mature state estimation strategies, such as inertial or vision-based methods, has difficulty obtaining satisfactory sensing performance on flapping wing platforms. Inertial sensors offer high sampling frequency but suffer from flapping-introduced oscillation and drift. External visual sensors, such as motion capture systems, can provide accurate feedback but come with a relatively low sampling rate and severe delay. This work proposes a novel state estimation framework to combine the merits from both to address such key sensing challenges of a special flapping wing platform—micro flapping wing rotors (FWRs). In particular, a cross-fusion scheme, which integrates two alternately updated Extended Kalman Filters based on a convex combination, is proposed to tightly fuse both onboard inertial and external visual information. Such a design leverages both the high sampling rate of the inertial feedback and the accuracy of the external vision-based feedback. To address the sensing delay of the visual feedback, a ring buffer is designed to cache historical states for online drift compensation. Experimental validations have been conducted on two sophisticated microFWRs with different actuation and control principles. Both of them show realtime and drift-free state estimation.

**Keywords:** microaerial vehicle; flapping wing rotorcraft; state estimation; sensor fusion

## 1. Introduction

Flapping wing Micro Aerial Vehicles (MAVs) adopt the flight principle of flying creatures and, thus, are promising for resembling animal-like extraordinary aerodynamic feats. To date, with the understanding of flapping flight aerodynamics and control strategies, the current flapping wing MAVs are becoming increasingly agile and miniaturized [1–11].

Among them, a new type of flapping-wing aircraft—micro Flapping Wing Rotorcrafts (FWRs)—integrates both the advantages of flapping and rotary wings, demonstrating superior capability of lift generation [1]. The state-of-the-art microFWRs even showcase several millimeter/milligram-scale designs [1,9,12–19] that have rarely been achieved by conventional fixed or rotary-winged vehicles. Such microFWRs are foreseen as alternatives to commercial drones and would be in used in increasing breadths of applications such as search and rescue, surveying and inspection, and aerial photography [20]. On the other hand, the small size and lightweight design of microFWRs result in undesired high control sensitivity, which limits the flight control performance significantly [2–4,6]. To address such a unique challenge, fast and accurate state estimation is the key prerequisite.

In order to accommodate stringent size and weight constraints, the microelectromechanical Systems (MEMSs)-based inertial measurement unit (IMU) is a practical sensing solution for miniaturized vehicles with high sampling rates (up to KiloHertz). To review, IMU accompanied with adequate state estimation algorithms works properly on conventional aircraft, especially on large-scale ones [21,22]. However, the direct adoption of such mature sensing solutions on flapping wing vehicles [2,3,10,11,23], usually results in inadequate performance due to their unsteady aerodynamic principles and time-varying system dynamics. In particular, the unsteady aerodynamic loading from the wings could induce severe vibration in inertial sensor readings [2,23–25], resulting in unmanageable sensing drift. As a result, flapping wing caused high-frequency varying aerodynamic loading lowers the accuracy of the IMU-based state estimation and affects flight control performance accordingly [2,11,23–25]. Vision-based motion capture system could be an effective alternative, which captures vehicle states by visual cues [26,27]. Nevertheless, the resolution of such external visual sensors may limit its tracking performance as the object becomes agile and tiny and environmental disturbances present (e.g., refractions or marker shielded). Most importantly, vision-based sensing runs on relatively low sampling frequencies (e.g., around 100–200 Hz for OptiTrack [26]), and it is hard to ensure real-time performance due to the slow image processing; namely, it always generates sluggish and delayed feedback, which could degrade control performance, if not losing the stability [26–28]. In fact, even for flying animals, the delayed sensory system can affect their flight control severely [29,30].

Due to the vibration caused by the particular flapping motion and the stringent constraints of scale, weight, and power, state estimation divergence becomes an open challenge for flapping wing platforms' long-term stable flight. As a result, using either IMU or external visual sensors alone cannot provide high-frequency and high-fidelity state estimation for microFWRs' flight control. It is desired to leverage both advantages of the above-mentioned sensing methods to obtain real-time and drift-free state estimation for precise control. To this end, particular sensor fusion challenges need to be addressed to attain satisfactory updating frequency, sensing accuracy, and delay compensation.

In this work, a state estimation framework is proposed to integrate multiple sensor readings, i.e., inertial and external visual sensors, to generate real-time and accurate state feedback for flapping wing MAVs. Because the sensing principles and readings of these two sensors are completely different, based on the convex combination theory, two particular Extended Kalman Filters (EKFs) are designed for sensor fusion. In order to enhance the computational efficiency and compensate for the sensing delay, a cross fusion framework is proposed to integrate these two EKFs' estimates, aiming to leverage both the high sampling rate of the inertial feedback and the accuracy of the visual feedback. A ring buffer is implemented to cache the historical state update to enable backtracking during cross fusion, which plays an important role in sensing delay compensation. The detailed workflow of the proposed state estimation framework is presented in Section 4. The proposed state estimation framework has been validated experimentally on two microFWRs with different actuation principles. As a result, the proposed state estimation method resembles the accuracy of the visual feedback and without delay. Meanwhile, it retains detailed flight state variations captured by the inertial sensor, demonstrating high-sensing bandwidths. During the bench tests, the updating frequencies of the proposed method are on par with the inertial sensors, but they are not limited to those sensors.

Most of the existing flapping-wing microvehicles use IMU or external camera as their sensory system [1–11]. Compared with their existing state estimation methods [2,11,23–28], the proposed method is able to take both high-fidelity and high-frequency state estimation results into account. Such performances and the robustness of the proposed method have been validated by two sophisticated microFWRs' real-world flight tests, which have even more complex aerodynamics than traditional flappers. We summarize the contributions as follows:

1. We proposed a generic method integrating inertial and external visual sensors by using EKFs' convex combination that simultaneously guarantees the accuracy and updating frequency of FWRs' state estimation. Such a method effectively addressed the above-mentioned sensing challenges of typical flapping-wing microvehicles;
2. A cross fusion framework to fusion pose information from the external visual sensors with the consideration of the transmission delay. This framework fundamentally benefits the control of small-sized agile aerial vehicles, which have high system sensitivity and were severely affected by the delay of pose feedback;
3. We implement the proposed method into two different prototypes of FWRs and conduct extensive real-world evaluation of our proposed method. Based on the test results, in addition to the aforementioned advantages, such a framework is capable of attenuating the influence of anomalous data.

The rest of the article is organized as follows. Sections 2 and 3 introduce the test platforms and the corresponding sensing challenges. Section 4 details the architecture and the algorithm of the proposed state estimation framework. Section 5 presents the experimental validation of the proposed state estimation framework. Section 6 summarizes this work.

## 2. Test Platforms and Their Sensory System

In order to validate the effectiveness of the proposed state estimation framework, two FWR platforms with their respective scales have been tested in this study. As shown in Figure 1, the test platforms come with different actuation principles, system parameters, control logic, and sensing coefficients. In this section, the details of such platforms are introduced below. The parametric comparison is summarized in Table 1.

**Table 1.** Wing parameters, mass, and inertia of the test platforms.

| Test Platform | MicroFWR (a) | MicroFWR (b) |
|:---:|:---:|:---:|
| Vehicle Parameters | | |
| Wing length ($R_w$) | 120 mm | 85 mm |
| Wingbeat frequency ($f$) | 16 Hz | 31 Hz |
| Total weight ($m$) | 27 g | 12.5 g |
| $x$-axis moments of inertia ($J_{xx}$) | 70,399 gmm$^2$ | 4238.13 gmm$^2$ |
| $y$-axis moments of inertia ($J_{yy}$) | 68,782 gmm$^2$ | 3970.16 gmm$^2$ |
| $z$-axis moments of inertia ($J_{zz}$) | 29,056 gmm$^2$ | 2440.95 gmm$^2$ |
| Sensor Specifications | | |
| IMU sampling rate | 512 | 1024 Hz |
| Gyroscope measurement range | $\pm2000$ deg/s | $\pm2000$ deg/s |
| Accelerometer measurement range | $\pm16$ g | $\pm16$ g |
| Vision feedback frequency | 100 Hz | 120 Hz |

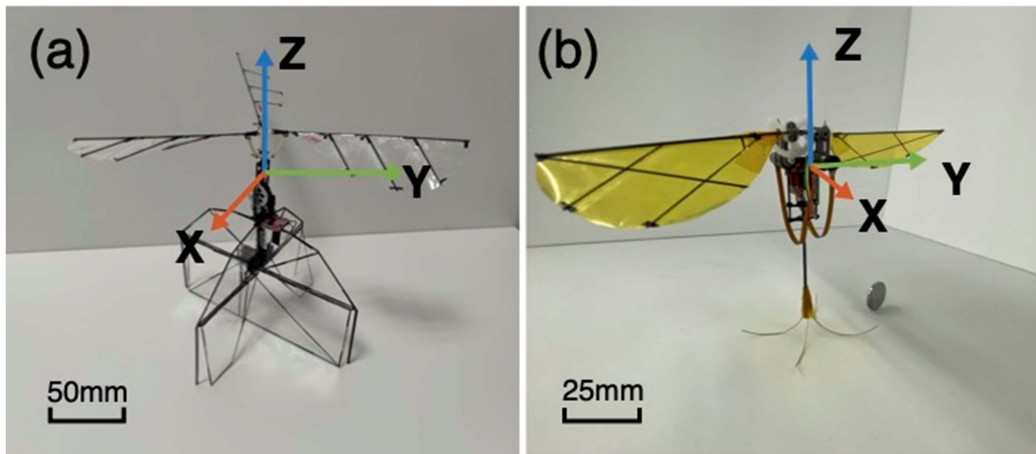

**Figure 1.** Illustration of test platforms: (**a**) a linkage-drive three-wing microFWR; (**b**) a motor direct-drive twin-wing microFWR.

### 2.1. Platform (a): A Linkage-Drive MicroFWR

The test platform microFWR (a) is driven by a four-bar-like linkage. Such a mechanism, as shown in Figure 2, drives three flapping wings to generate rotation torque and lift. The flapping amplitude is constrained by the linkage while the flapping frequency can be altered for lift control. Since the constrained wing trajectory is not able to generate control torque, the control surfaces have been adopted on the tail to generate control torque for flight control. The onboard electronics include a STM32F405 microcontroller, an MPU9250 IMU sensor, motor and servo drivers, and detachable wireless telemetry. Among them, the IMU sensor provides high-frequency inertia feedback, and telemetry is used to receive external visual feedback. Detailed design of FWR (a) is presented in [10].

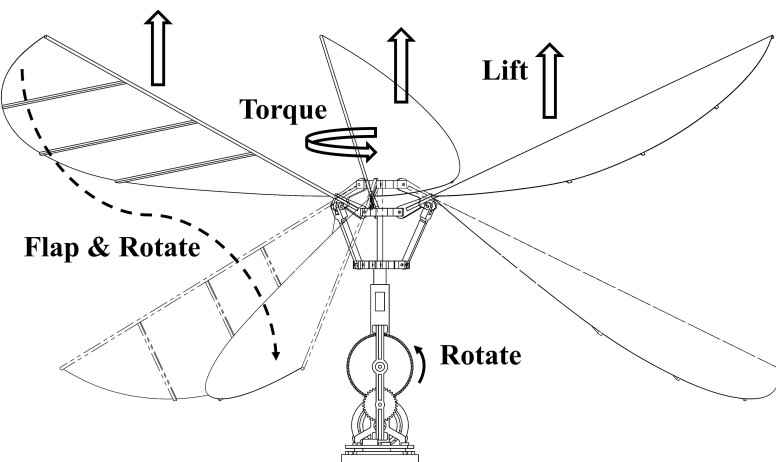

**Figure 2.** Driving principle of FWR (a).

### 2.2. Platform (b): A Motor Direct-Drive MicroFWR

Test platform FWR (b) is directly actuated by two bi-directional rotating brushless dc motors; thus, it avoids the use of the complicated transmission system, as shown in platform (a). Its prototype is shown in Figure 3. For such a design, each wing is driven by its paired dc motor independently, which is similar to the Robotic Hummingbird designed by Tu et al. [31], but two wings are mounted anti-symmetrically. Therefore, the aerodynamic principle of FWR (b) is more similar to normal birds rather than hummingbirds. The motor equips Hall-sensor feedback for commutation control, yielding bi-directional rotation to enable reciprocating wing motion. Reduction gears and torsional springs are installed to connect the motor and wing for torque transmission. With aerodynamic and inertial

loading, the wing is designed to rotate passively. Wing kinematics can be controlled by modulating the input voltage of the motor. The discrepancy of the wing kinematics can generate control torques to stabilize the vehicle.

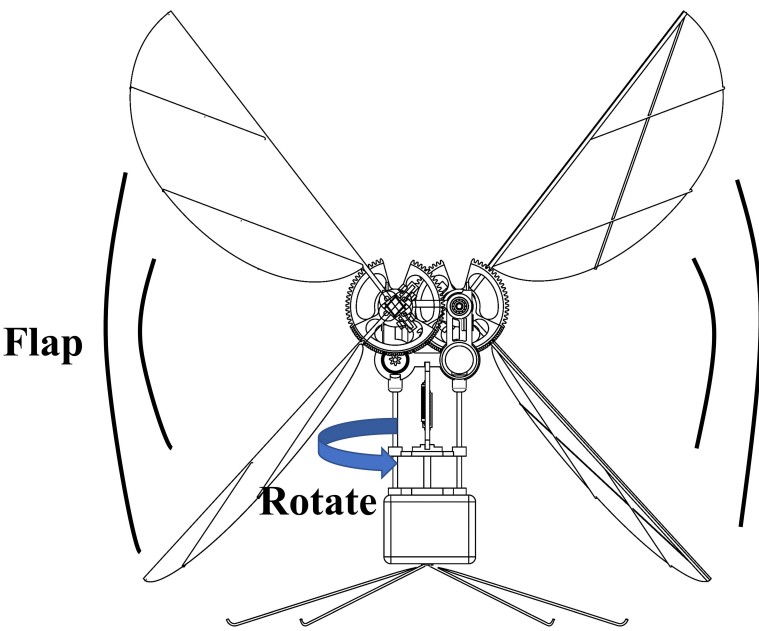

**Figure 3.** Driving principle of FWR (b).

### 2.3. Sensory System

The sensory system used in this study mainly consists of onboard IMU and offboard visual sensors. The specific onboard IMU is MPU9250, which contains a three-axis gyroscope, a three-axis accelerometer, and a three-axis magnetometer. Its updating frequency reaches as high as 400 kHz, which is sufficient for high bandwidth system control. For external visual sensing, we used OptiTrack (https://OptiTrack.com (accessed on 7 March 2022))—a motion capture system that relies on multiple infrared cameras to track the markers dotted on the test platform. The setup is depicted in Figure 4. Note, different sensing frequencies of the two test platforms were implemented in order to verify the generality of the later proposed sensor fusion method. According to such sensory system setups, experimental comparative studies with all three different state estimation strategies, e.g., onboard IMU only, offboard OptiTrack only, and the proposed IMU-OptiTrack fusion, have been conducted. A sample result that demonstrates the performance discrepancy is shown and discussed in Section 5.

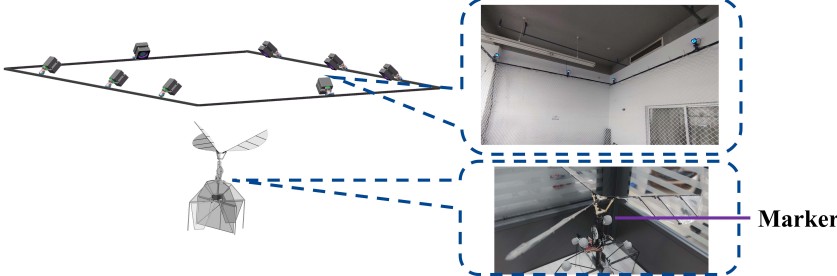

**Figure 4.** The setup of motion capture system.

## 3. State Estimation Challenges of FWRs

With the test platforms and sensor systems as described in Section 2, the exact state estimation challenges of microFWRs can be found systematically. Such challenges motivate this work accordingly. In this section, we introduce the respective limitations of IMU and external visual sensors on microFWRs.

### 3.1. Limitation of Inertial Sensors

MEMS-tech based IMUs can provide high-bandwidth inertial feedback, which is desirable for miniaturized aerial vehicles with stringent size, weight, and control sensitivity constraints. Although it works reasonably well on conventional vehicles with proper sensor fusion algorithms, simply implementing it on flapping wing vehicles usually obtains poor performance. Flapping-wing vehicles are known to face severe body vibration due to the high-frequency reciprocal wing motion and complex time-varying aerodynamics [2,23–25]. In particular, such severe oscillation not only generates undesired oscillatory control error but also affects IMU readings significantly, resulting in untrusted sensor feedback, as shown in Figure 5. Based on the previous study [2,23–25], such severe vibration is prominent in accelerometer measurements during flight. In addition, the gyroscope also demonstrates unmanageable sensor drift due to the bias and noise uncertainty. Without reliable accelerometer readings, merely using a gyroscope cannot sustain accuracies for long-term estimation.

Taking FWR (b) as an example: With the raw IMU data shown in Figure 5, several mature sensor fusion solutions have been tested, including complimentary filter and Extended Kalman filter. Their respective best performance is shown in Figure 6. Although it is already the best performance, it cannot be applied to flight control due to such obvious state estimation errors. In order to verify the dilemma of using IMU on flapping wing systems, quantitative studies have been conducted. As a result, in the case of completely distrusting the accelerometer, the state estimation will quickly diverge. Nevertheless, continuously increasing the weight of acceleration information in sensor fusion will obviously result in greater estimation errors [24].

In fact, two platforms in this article show similar sensing issues as the wing starts flapping. Relying on IMU alone renders it hard to achieve long-term reliable state estimation.

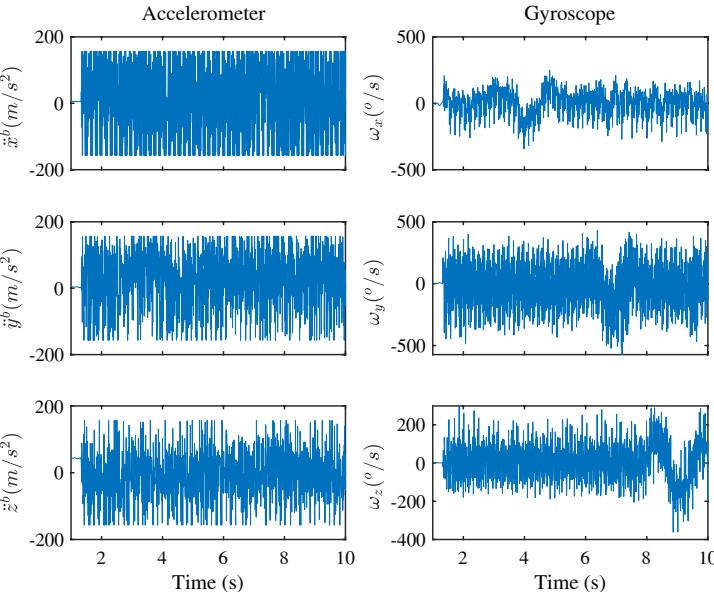

**Figure 5.** The raw IMU data of FWR (b).

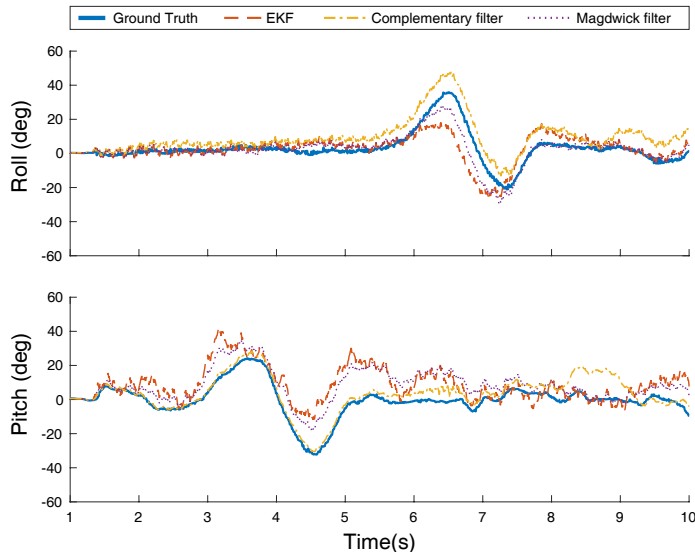

**Figure 6.** Several mature sensor fusion solutions of FWR (b) IMU data.

### 3.2. Limitation of External Visual Sensors

In order to avoid such particular challenges on IMU readings, an external motion capture system could be a practical method. Such a system relies on several infrared cameras to track predefined objects in real-time. Since it needs to process all camera information, the processed data are updated slowly and with certain delay, similarly to other visual-based sensing approaches.

The visual feedback delay can be determined by conducting a delay calibration test. During the test, we change the coordinate of the tracking object instantly by switching the lighting sequence of infrared LED1 and LED2. The delay can be determined by synchronizing the LED switching command and state feedback change in the time sequence, as shown in Figure 7. Three different data transmission schemes were implemented and tested: wired serial communication, 2.4 GHz wireless module nRF24L01 transmission, and ESP8266 WiFi transmission. The cable length of the serial communication is about 3 m. A long cable length was implemented to prevent affecting the free flight performance of FWR. The calibration result is shown in Figure 8, the result demonstrated that the wired serial communication has the lowest latency in about 10 ms, and the latency of the ESP8266 WiFi module's transmission is slightly higher than serial communication in about 20 ms, while the latency of nRF24L01 transmission is the highest and has bad consistency. During the flight experiment conducted in Section 5, the wired serial was used to provide stable and reliable pose feedback.

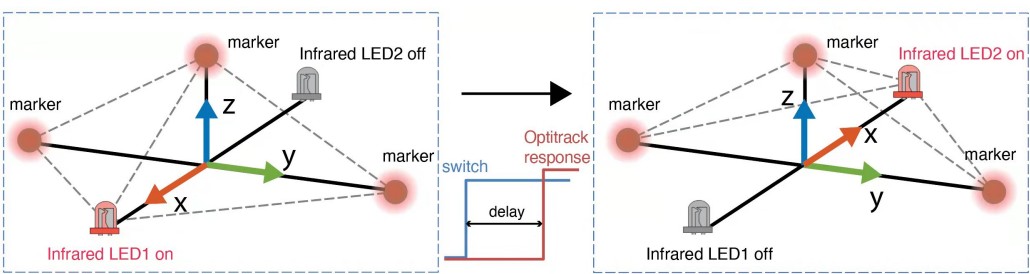

**Figure 7.** The delay calibration setup.

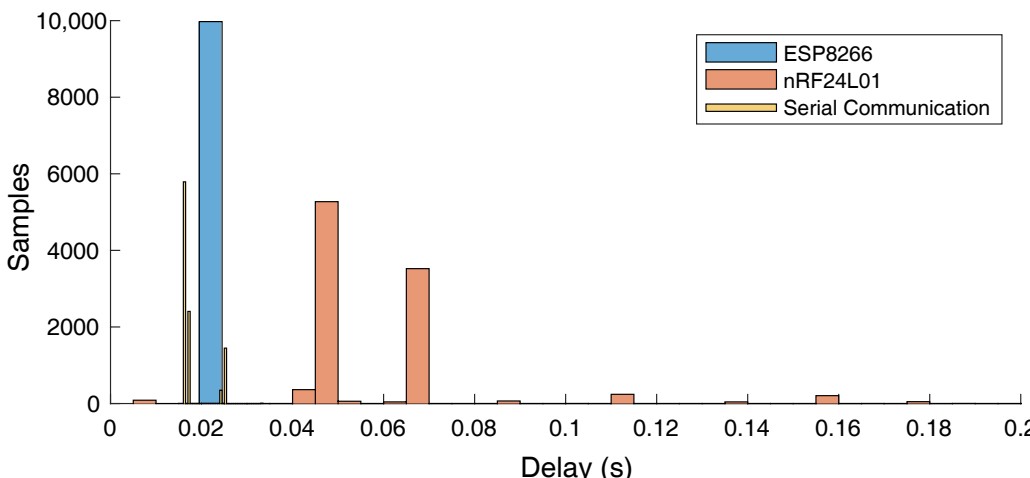

**Figure 8.** Delay calibration result.

## 4. State Estimation Framework

In order to address the specific sensing challenges obtained in Section 3, we propose a state estimation method in this section that can integrate two different sensors properly and provide fast and accurate state feedback for flight control. In this section, we first define the coordinates and vehicle states used in this study. Then, an EKF-based cross fusion framework is introduced in detail.

### 4.1. Spatial Frames

The spatial frames involved in our system consist of the following:

1.  Vehicle body frame: Vehicle body frame is attached to the Center of the Gravity (CoG) of the vehicle and denoted by $\bullet^b$;
2.  Onboard sensor frame: Onboard IMU sensor frame is a local frame in which it generates 10-DoF inertial feedback of the vehicle, including three-axis acceleration $a^b$, three-axis angular rate $\omega^b$, three-axis magnetic field $m^b$, and air-pressure. In this study, we attach the IMU frame to the CoG of the test vehicle and mark it as our estimated body frame $\bullet^b$;
3.  Inertial frame: As shown in Figure 9, the conventional frame is introduced in which the external visual-feedback system operates as the inertial frame. The origin of the Inertial frame is arbitrarily set, which is defined by the vision system's calibration. The z-axis is often chosen to be orthogonal to the local ground plane.

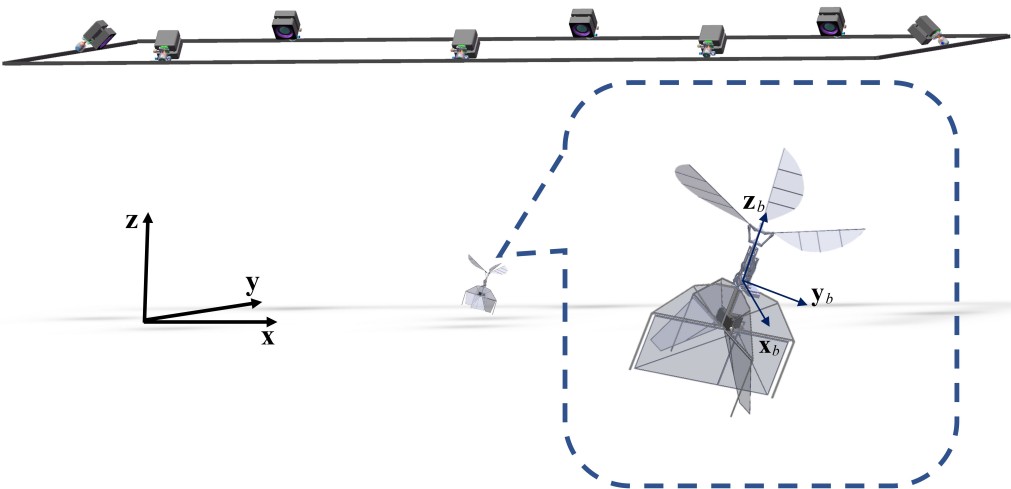

**Figure 9.** The Inertial and body frame of FWR.

### 4.2. Vehicle States

A generic rigid body assumption is introduced for the state prediction of general MAVs, model physical parameters such as mass and inertia are assumed to be constant. The dynamic model follows the Newton–Euler equation, shown in Equation (1):

$$
\begin{aligned}
\dot{\mathbf{p}} &= \mathbf{v} \\
m\ddot{\mathbf{p}} &= \mathbf{R}\mathbf{f}^b + mg\mathbf{e}_3 \\
\dot{\mathbf{R}} &= \mathbf{R}\hat{\boldsymbol{\omega}}^b \\
\mathbf{J}\dot{\boldsymbol{\omega}}^b + \boldsymbol{\omega}^b \times \mathbf{J}\boldsymbol{\omega}^b &= \boldsymbol{\tau}^b
\end{aligned}
\tag{1}
$$

where $\mathbf{p} = [x, y, z]^T$ and $\mathbf{v} = \dot{\mathbf{p}}$ are the vehicle's position and velocity in the inertial frame, $\mathbf{R} \in \mathbb{R}^{3\times3}$ is the rotation matrix, $\mathbf{f}^b = [F_x, F_y, F_z]^T$ is the control force applied on the vehicle, $\mathbf{e}_3$ is a unit vector $[0, 0, 1]^T$, $\boldsymbol{\omega}^b = [\omega_x, \omega_y, \omega_z]^T$ is the body angular velocity, and $\hat{\boldsymbol{\omega}}^b$ is the skew-symmetric matrix of vector $\boldsymbol{\omega}^b$, as shown in Equation (2). $\mathbf{J} \in \mathbb{R}^{3\times3}$ is the inertia matrix, and $\boldsymbol{\tau}^b = [\tau_x, \tau_y, \tau_z]^T$ is the control torque applied on the vehicle.

$$
\hat{\boldsymbol{\omega}}^b = \begin{bmatrix} 0 & -\omega_z & \omega_y \\ \omega_z & 0 & -\omega_x \\ -\omega_y & \omega_x & 0 \end{bmatrix}
\tag{2}
$$

A quaternion based rotation matrix $\mathbf{R}(\mathbf{q})$ is defined by the following (3):

$$
\mathbf{R}(\mathbf{q}_k) = \begin{bmatrix} q_1^2 + q_2^2 - q_3^2 - q_4^2 & 2(q_2q_3 - q_1q_4) & 2(q_1q_3 + q_2q_4) \\ 2(q_2q_3 + q_1q_4) & q_1^2 - q_2^2 + q_3^2 - q_4^2 & 2(q_3q_4 - q_1q_2) \\ 2(q_2q_4 - q_1q_3) & 2(q_1q_2 + q_3q_4) & q_1^2 - q_2^2 - q_3^2 + q_4^2 \end{bmatrix}
\tag{3}
$$

where $\mathbf{q} = [q_1, q_2, q_3, q_4]$ is the quaternion vector. The transition between Euler angle $[\phi, \theta, \psi]$ and $\mathbf{q}$ is given by Equation (4).

$$
\begin{bmatrix} \phi \\ \theta \\ \psi \end{bmatrix} = \begin{bmatrix} \arctan \frac{2(q_0q_1 + q_2q_3)}{1 - 2(q_1^2 + q_2^2)} \\ \arcsin(2(q_0q_2 - q_3q_1)) \\ \arctan \frac{2(q_0q_3 + q_1q_2)}{1 - 2(q_2^2 + q_3^2)} \end{bmatrix}
\tag{4}
$$

Moreover, the derivative of the quaternion is calculated using Equation (5).

$$
\dot{\mathbf{q}} = \frac{1}{2}\mathbf{q} \otimes [0, \omega_x, \omega_y, \omega_z]^T = \begin{bmatrix} -q_2\omega_x - q_3\omega_y - q_4\omega_z \\ q_1\omega_x - q_4\omega_y + q_3\omega_z \\ q_4\omega_x + q_1\omega_y - q_2\omega_z \\ -q_3\omega_x + q_2\omega_y + q_1\omega_z \end{bmatrix}
\tag{5}
$$

Based upon it, the vehicle state is defined by the following:

$$
\hat{\mathbf{x}} = [\mathbf{p}, \mathbf{v}, \mathbf{q}, \boldsymbol{\omega}^b]
\tag{6}
$$

by using simple system identification, these system parameters can be easily obtained.

### 4.3. State Prediction

The generic system model can be written into the following discrete form as Equation (7):

$$
\begin{aligned}
\mathbf{x}_{k+1} &= \mathbf{f}(\mathbf{x}_k, \mathbf{u}_k, \mathbf{b}_k) + \mathbf{w}_k, \\
\mathbf{y}_{k+1} &= \mathbf{H}\mathbf{x}_k + \mathbf{v}_k.
\end{aligned}
\tag{7}
$$

where $\mathbf{x} = [x, y, z, \dot{x}, \dot{y}, \dot{z}, q_1, q_2, q_3, q_4, \omega_x, \omega_y, \omega_z]^T$ is the state variables vector, the control vector is $\mathbf{u} = [u_{trust}, u_{roll}, u_{pitch}, u_{yaw}]^T$, $\mathbf{b}$ is the constant sensing bias, $\mathbf{y}$ is the output variables vector, $\mathbf{v}$ and $\mathbf{w}$ are the zero-mean uncorrelated Gaussian noise, and $k$ represents the discrete time step. $\mathbf{H}$ is observation matrix.

Analogous to Kalman filter, a discrete state prediction is given by Equation (8).

$$
\begin{aligned}
\mathbf{p}_{k+1} &= \mathbf{p}_k + \dot{\mathbf{p}}_k \Delta t \\
\dot{\mathbf{p}}_{k+1} &= \dot{\mathbf{p}}_k + \frac{\mathbf{R}(\mathbf{q}_k)[0, 0, F_L]^T \Delta t}{m} \\
\mathbf{q}_{k+1} &= \mathbf{q}_k + \dot{\mathbf{q}}_k \Delta t \\
\boldsymbol{\omega}_{k+1} &= \boldsymbol{\omega}_k + \dot{\boldsymbol{\omega}}_k \Delta t \\
\mathbf{b}_{k+1} &= \mathbf{b}_k
\end{aligned}
\tag{8}
$$

Here, $F_L$ is the body force, which is the function of input trust signal $u_{trust}$, and $m$ is the mass of microFWR.

Angular acceleration can be derived from body dynamics using Equation (9).

$$
\dot{\boldsymbol{\omega}} = \begin{bmatrix} \frac{I_y - I_z}{I_x} \omega_y \omega_z + \frac{T_x}{I_x} \\ \frac{I_x - I_z}{I_y} \omega_x \omega_z + \frac{T_y}{I_y} \\ \frac{I_x - I_y}{I_z} \omega_x \omega_y + \frac{T_z}{I_z} \end{bmatrix}
\tag{9}
$$

Here, $T_x, T_y, T_z$ are the three-axis torques in terms of the attitude control input $u_{roll}, u_{pitch}, u_{roll}$ [10].

### 4.4. Convex Combination Based Sensor Fusion

The estimation of the same flight state by two different sensors can be formulated as a convex combination problem [32,33]. As an important component of the proposed state estimation framework, EKF can be treated as a recursive form of Gauss–Newton optimization on a typical Kalman filter [34]. The key is to update the reference to address the significant nonlinearity during filtering. In this study, we combine two EKF filters following a cross-fusion law. The mixed sensing result emphasizes the qualities and overcomes the defects of each used sensor.

In particular, the data from IMU and OptiTrack update at different rates. Thus, two sets of EKF methods are implemented to estimate the state of tested FWR. IMU data are available at high sampling rates. The sensor fusion is running at the same frequency as the IMU updating. The a priori estimation of the state of FWR is given by Equation (10).

$$
\begin{aligned}
\hat{\mathbf{x}}_{k|k-1} &= \mathbf{f}\left(\hat{\mathbf{x}}_{k-1|k-1}, \mathbf{u_k}\right) \\
\mathbf{P}_{k|k-1} &= \mathbf{F}_k \mathbf{P}_{k-1|k-1} \mathbf{F}_k^T + \mathbf{Q}_k
\end{aligned}
\tag{10}
$$

Here, $\hat{\bullet}$ is the estimated variable, $\mathbf{P}_{k|k-1}$ is the a priori error covariance matrix, and $\mathbf{F}_k$ is the derivative of $\mathbf{f}(\mathbf{x}, \mathbf{u})$ at $\mathbf{x}_k$. Then, the measurement vector and Kalman gain $\mathbf{K}_k^n$ are updated by using Equation (11).

$$
\begin{aligned}
\hat{\mathbf{Y}}_k^n &= \mathbf{H}_k^n \hat{\mathbf{x}}_{\mathbf{k|k-1}} \\
\mathbf{S}_k^n &= \mathbf{H}_k^n \mathbf{P}_{k-1|k-1} \mathbf{H}_k^{n^T} + \mathbf{R}_k^n \\
\mathbf{K}_k^n &= \mathbf{P}_{k|k-1} \mathbf{H}_k^{n^T} \mathbf{S}_k^{n-1} \\
n &\in \{1, 2\}
\end{aligned}
\tag{11}
$$

Here, $\mathbf{Q}_k$ and $\mathbf{R}_k^n$ are the covariance matrices of the noises of $\mathbf{v}$ and $\mathbf{w}$ in Equation (7), $\mathbf{S}_k^n$ is the observation error covariance matrix, and $\mathbf{I}$ is the identity matrix and. When IMU data are available and no OptiTrack data were updated, we have $n = n_1$, and the observation

of the body angular rate $\mathbf{Y}_k^1 = [g_x, g_y, g_z]^T$ is obtained from the gyroscope. When a new OptiTrack data frame is received, we have $n = n_2$, and $\mathbf{Y}_k^2 = [x, y, z, q_1, q_2, q_3, q_4]^T$; the corresponding observation matrices $\mathbf{H}_k^1$ and $\mathbf{H}_k^2$ and the observation covariance matrices $\mathbf{R}_k^1$ and $\mathbf{R}_k^2$ are provided by Equation (12).

$$
\mathbf{H}_k^1 = \begin{bmatrix} \mathbf{0}_{10\times10} & \mathbf{0}_{10\times3} \\ \mathbf{0}_{3\times10} & \mathbf{I}_{3\times3} \end{bmatrix} \mathbf{H}_k^2 = \begin{bmatrix} \mathbf{I}_{3\times3} & \mathbf{0}_{3\times3} & \mathbf{0}_{3\times4} & \mathbf{0}_{3\times3} \\ \mathbf{0}_{3\times3} & \mathbf{0}_{3\times3} & \mathbf{0}_{3\times4} & \mathbf{0}_{3\times3} \\ \mathbf{0}_{4\times3} & \mathbf{0}_{4\times3} & \mathbf{I}_{4\times4} & \mathbf{0}_{4\times3} \\ \mathbf{0}_{3\times3} & \mathbf{0}_{3\times3} & \mathbf{0}_{3\times4} & \mathbf{0}_{3\times3} \end{bmatrix},
$$

$$
\mathbf{R}_k^1 = \begin{bmatrix} \mathbf{0}_{10\times10} & \mathbf{0}_{10\times3} \\ \mathbf{0}_{3\times10} & 10^{-3} \times \mathbf{I}_{3\times3} \end{bmatrix} \mathbf{R}_k^2 = \begin{bmatrix} 10^{-6} \times \mathbf{I}_{3\times3} & \mathbf{0}_{3\times3} & \mathbf{0}_{3\times4} & \mathbf{0}_{3\times3} \\ \mathbf{0}_{3\times3} & \mathbf{0}_{3\times3} & \mathbf{0}_{3\times4} & \mathbf{0}_{3\times3} \\ \mathbf{0}_{4\times3} & \mathbf{0}_{4\times3} & 10^{-5} \times \mathbf{I}_{4\times4} & \mathbf{0}_{4\times3} \\ \mathbf{0}_{3\times3} & \mathbf{0}_{3\times3} & \mathbf{0}_{3\times4} & \mathbf{0}_{3\times3} \end{bmatrix} \quad (12)
$$

$$
\mathbf{Q}_k = \begin{bmatrix} 10^{-1} \times \mathbf{I}_{3\times3} & \mathbf{0}_{3\times3} & \mathbf{0}_{3\times4} & \mathbf{0}_{3\times3} \\ \mathbf{0}_{3\times3} & \mathbf{0}_{3\times3} & \mathbf{0}_{3\times4} & \mathbf{0}_{3\times3} \\ \mathbf{0}_{4\times3} & \mathbf{0}_{4\times3} & 10^{-2} \times \mathbf{I}_{4\times4} & \mathbf{0}_{4\times3} \\ \mathbf{0}_{3\times3} & \mathbf{0}_{3\times3} & \mathbf{0}_{3\times4} & 10^{-1} \times \mathbf{I}_{3\times3} \end{bmatrix}
$$

As the data fusion result may come from two different sensors, the state estimation using only the measurement information and the a posteriori error covariance matrix from the sensor $n$ at time $k$ is given by Equation (13).

$$
\widehat{\mathbf{x}}_{k|k}^n = \widehat{\mathbf{x}}_{k|k-1} + \mathbf{K}_k^n \left( \mathbf{Y}_k^n - \widehat{\mathbf{Y}}_k^n \right)
$$
$$
\mathbf{P}_{k|k}^n = (\mathbf{I} - \mathbf{K}_k^n \mathbf{H}_k^n) \mathbf{P}_{k|k-1}^n \quad (13)
$$
$$
n \in \{1, 2\}
$$

The estimation error is given by Equation (14).

$$
\mathbf{x}_{k|k}^n = \mathbf{x} - \widehat{\mathbf{x}}_{k|k}^n \quad (14)
$$

We are given the influence of the possible correlation between the local estimation errors. The local estimation errors of any two sensors are correlated. This kind of correlation should be considered when performing data fusion. The cross-covariance between the local estimation errors of the sensors is provided by Equation (15).

$$
\mathbf{P}_{k|k} \triangleq E\left[ \mathbf{x}_{k|k}^n \left( \mathbf{x}_{k|k}^n \right)^T \right] = \mathbf{P}_{k|k}^1 \mathbf{P}_{k|k-1}^{-1} \mathbf{P}_{k|k}^2 \quad (15)
$$

Based on the above derivation, the proposed convex combination based algorithm takes into account the correlation between the estimation errors of each sensor, which is the key advantage. Nevertheless, this method needs the gain of the filter, and the historical measurement matrix needs to be returned to the step where the last time cross fusion was completed. Such a method requires a lot of onboard calculation and storage space to iteratively calculate the covariance matrix between the estimation errors of the various sensors, which may lower its sensing efficiency significantly.

### 4.5. Cross Fusion Framework

In order to boost sensing efficiency, we propose a cross fusion framework, which is based on the systematic consideration of the characteristics of the two airborne sensors. Due to the high sensing frequency of IMU, we use OptiTrack update as the keyframe for sensor fusion. With the known OptiTrack delay step, we can trace back to the time step corresponding to the received OptiTrack data.

Base on the aforementioned state estimation methods, a cross-updating law is proposed to fuse the respective inertial and visual readings with different frequencies and states. Such a framework is illustrated in Figure 10.

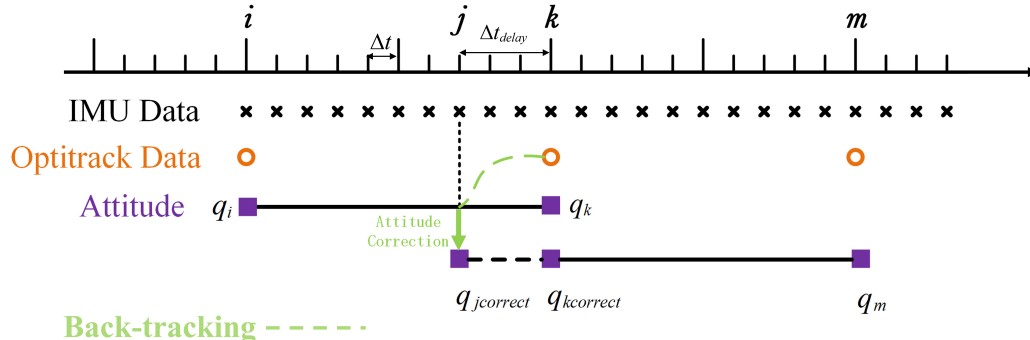

**Figure 10.** Framework of proposed method.

In the framework, the inertial readings are available at a high updating frequency up to kilohertz. Therefore, it dominates the maximum state estimation frequency that coordinates with the flight control needs. In addition to the updating frequency, accuracy is also important. The sluggish external visual feedback plays an important role to guarantee estimation accuracy, which initiates an online calibration to refresh cached estimates in the buffer queue.

At a sample frame as shown in Algorithm 1, once the inertial sensor is available while visual reading does not update, the vehicle relies on it to perform state estimation. Meanwhile, the state will be cached in the buffer. The buffer size is mainly determined by visual feedback frequency. It should cover enough historical frames to integrate the delayed visual feedback. When a new visual feedback frame is received, it would trace back to the cached states to find the best match and instantiate a new state object. This object could replace the original state in the queue and trigger a rectification of the rest states, which would be refreshed by the cached inertial measurements to ensure fast convergence to the most recent state. Such a cross-updating law is able to eliminate sensing drifts as well as sensing delays.

To further boost computation efficiency, introducing a delay compensation factor to interpret sensor delay in the algorithm is suggested, which can be obtained by using a simple calibration test or by manually tuning. This factor determines a certain backtracking step to omit the computationally burdensome online matching process. For example, with a known delay, we can define the back steps in the cached states queue corresponding to the received visual data. After retrieving, it is used to update the initial frame and enable rectification until the most recent state feedback is ready. Meanwhile, as it has relatively low computation efficiency for some microcontroller, an IMU pre-integration algorithm [35] is suggested in the forward-calibration step. As shown in Figure 10, after the backtracking step, the correct attitude in time $j$ is acquired; then, we can calculate the relative attitude between time $j$ and $k$ using Equation (16).

$$\mathbf{q}_{kcorrect} = \mathbf{q}_{jcorrect} \otimes \mathbf{q}_{jk}$$
$$\mathbf{q}_{jk} = \int_{t \in [j,k]} \mathbf{q}_{jt} \otimes \begin{bmatrix} 0 \\ \frac{1}{2}\omega^t \end{bmatrix} \delta t \tag{16}$$

---

**Algorithm 1** Cross Fusion.

---

**Notation:** State $x$, History State $x_{cache}$, Imu $I$, History imu $I_{cache}$

**Output:** $x_{estimate}$

1: **while** True **do**
2:     step++
3:     **if** No OptiTrack Update **then**
4:         $x_{predict}$=**StatePredict()**
5:         $x_{estimate}$=**StateEstimateIMU()**
6:     **else**
7:         $j = step - \Delta t_{delay}$
8:         $x = x_{cache}(j)$
9:         $x_{predict}$=**StatePredict()**
10:        $x_{estimate}$=**StateEstimateOptitrack()**
11:        **for** $j = step - \Delta t_{delay} + 1$ to $step + 1$ **do**
12:           $I = I_{cache}(j)$
13:           $x_{predict}$=**StatePredict()**
14:           $x_{estimate}$=**StateEstimateIMU()**
15:           $x_{cache}(j) = x_{estimate}$
16:        **end for**
17:     **end if**
18:     $I_{cache}(step) = I$
19:     $x_{cache}(step) = x_{estimate}$
20: **end while**
21:
22: **return** $x_{estimate}$

---

## 5. Experimental Results

To validate the proposed method, we conduct real-world experiments on two different FWR platforms described in Section 2. During each flight test, three different sensor fusion methods—pure IMU-based EKF method, OptiTrack feedback, and the proposed method—were adopted and recorded to estimate the pose of the FWRs for comparison. To address this, due to the limited capture area of our OptiTrack systems, the sustained flight time of different test is inconsistent. As for comparison, we take 10 second of data in each free flight experiment.

### 5.1. Sensor Fusion on FWR (a)

As shown in Figure 11a, in this test, the sensory system is constructed by an onboard MPU9250 inertial sensor together with an offboard OptiTrack visual feedback. From Figure 8, such offboard visual feedback can be treated as ground truth with certain delays. The buffer stores 50 previous frames that cover 0.1 s historical information. It is necessary to improve the quality of sensors and readings by filtering in order to remove noise. A low-pass filter(LPF) is implemented here, which has a cut-off frequency of 50 Hz.

For this test platform, we focus on roll and pitch estimation, since the slight position drift and rotating yaw do not affect hover stability. Due to the reciprocal up–down motion of the flapping wing rotor, it generates significant z-axis vibration, which affects the accuracy of the inertial measurements and flight control. Consequently, roll angle estimation based on the merely inertial sensor gradually diverges, as shown in Figure 12c. The root mean square errors (RMSEs) of using individual IMU with Extended Kalman filter and the proposed fusion method are listed in Table 2. As shown in the zoomed area in Figure 12, the proposed method could obviously eliminate the visual feedback delay. In this case, the external vision-based sensor and the proposed framework both demonstrate their proper performance in tracking the motion of the test vehicle. Results show that the proposed method has the overall best capability to track ground truth without delay.

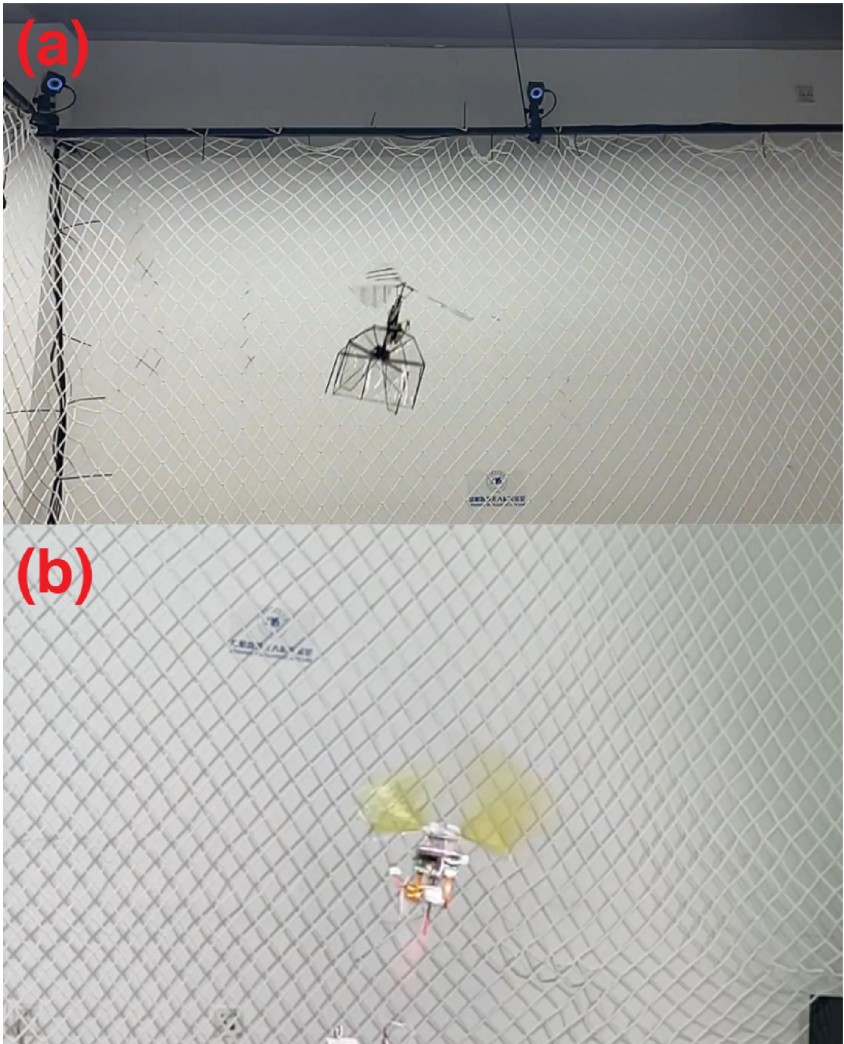

**Figure 11.** Sensor fusion experiments. (**a**) FWR (a); (**b**) FWR (b).

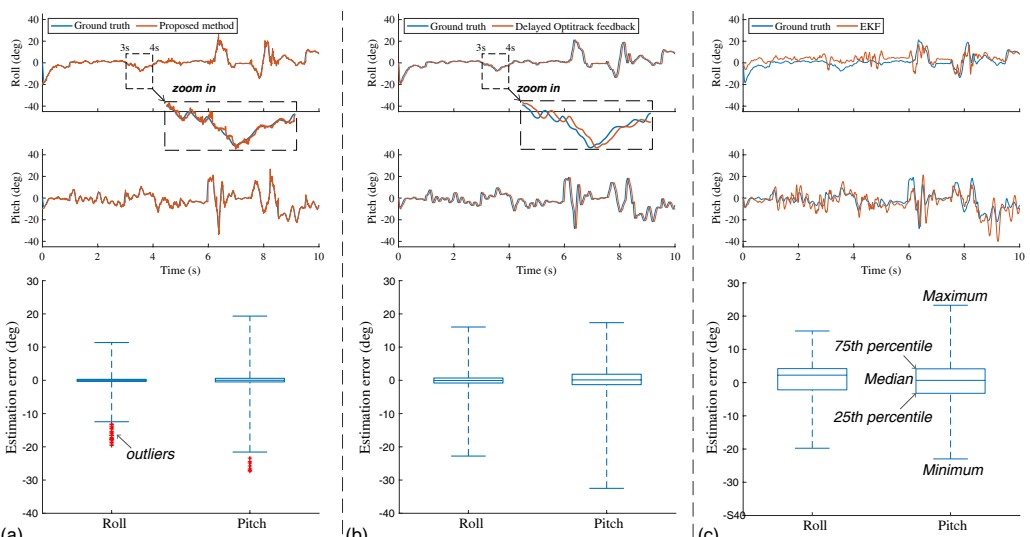

**Figure 12.** Attitude estimation result on FWR (a): (**a**) altitude estimation result using proposed method; (**b**) altitude feedback of delayed OptiTrack feedback; (**c**) altitude estimation result using EKF.

**Table 2.** RMSE of different fusion methods on FWR (a).

| Method | Roll | Pitch |
|---|---|---|
| Proposed Method | **1.8160** | **1.7444** |
| OptiTrack with delay | 3.4175 | 3.9250 |
| Extended Kalman Filter | 6.0504 | 5.8439 |

*5.2. Sensor Fusion on FWR (b)*

As shown in Figure 11b, in this test, we use FWR (b) as the test platform with the same buffer size for state estimation. Since it is much smaller and lighter than FWR (a), it will be correspondingly more sensitive to vibration and sensing delay.

Compared with the former test, this case is more challenging. Not only has the vehicle become smaller and more agile, but the system oscillation is also more severe. The high-frequency reciprocating wing motion can result in fierce vibration along the dorsal thorax direction of the test platform, causing inertial measurement drift. Therefore, the cut-off frequency of LPF was changed to 150Hz while the buffer size was the same as the former test.

As a result, IMU-based state estimation will cause obvious sensing bias, as shown in Figure 13c. The RMSE comparison result is shown in Table 3. The RMSE of the complementary filter-based state estimation is also tested. As a result, such inaccurate feedback can result in a quick divergence of flight control, causing serious consequences. Moreover, in this case, the visual feedback delay does affect control stability, since it is already greater than a wingbeat cycle. As the delay directly corresponds to the response latency of the flight control, the inconsistent control command can cause stability issues inevitably. Both EKF and complementary filter demonstrates poor performance. To improve state estimation performance, the proposed framework is implemented. The result is demonstrated in Figure 13, which generates reliable feedback without unforeseen drift and delay. Simultaneously, the proposed framework can also eliminate the outlier data from OptiTrack induced by the violent oscillation of the platform, as shown in the dashed box in Figure 13a,b. Furthermore, the updating frequency of the proposed method is 1 KHz, which is the same as the inertial readings.

The estimation result shows that the IMU-based method has high frequency, and the real time attitude of microFWRs has obvious bias. Compared between Tables 2 and 3, a worse result was generated when the platform was smaller, which indicates that severe oscillation will lead to the the unmanageable measurement noise of the accelerometer. OptiTrack feedback has accurate pose measurements, but they are affected by low sampling rate and transmission delay, violent vibration also results in some undesired outlier. To combine the advantage and take transmission delay into consideration, the proposed method provides state estimation with high frequency and accuracy.

**Table 3.** RMSE of different fusion method on FWR (b).

| Method | Roll | Pitch |
|---|---|---|
| Proposed Method | **5.3086** | **4.3171** |
| OptiTrack with delay | 6.1177 | 5.3715 |
| Extended Kalman filter | 11.1964 | 9.5309 |
| Complementary Filter | 10.8845 | 7.8824 |

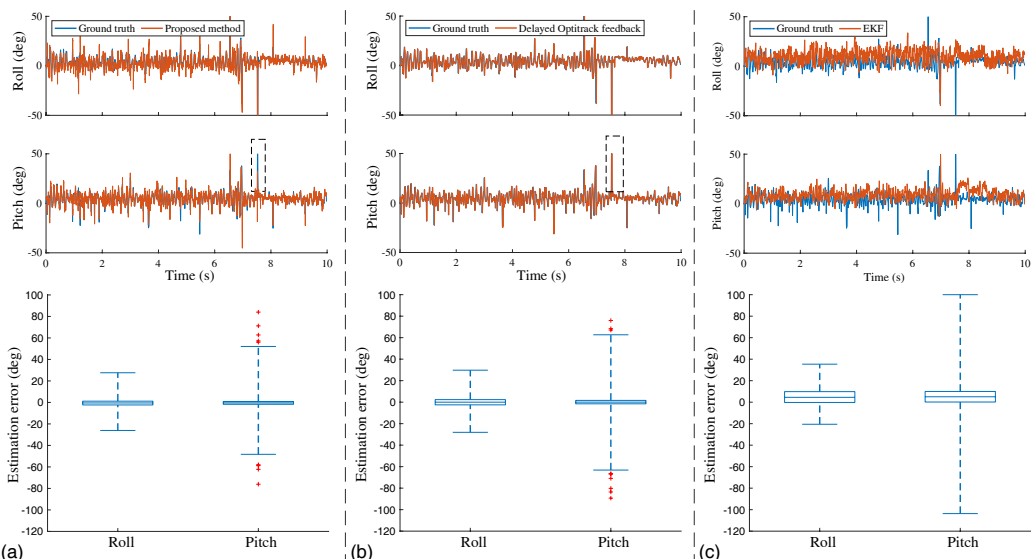

**Figure 13.** Attitude estimation result on FWR (b): (**a**) altitude estimation result using proposed method; (**b**) altitude feedback of delayed OptiTrack feedback; (**c**) altitude estimation result using EKF.

## 6. Conclusions

In this study, a generic state estimation solution was proposed for microFWRs. The contribution mainly lies in the cross fusion framework, which integrates the high sampling frequency of the inertial sensor and accurate visual feedback, yielding a real-time and drift-free estimation result. The framework has been experimentally validated on two sophisticated FWRs with different actuation principles. Both of them demonstrate high-frequency and high-fidelity sensing performance simultaneously. As observed from the results, the pure IMU-based state estimation can provide high frequency attitude with low latency, while the severe vibration of the FWRs' platform will result in high sensing bias. The pure visual feedback attitude state can provided the accurate pose of FWRs, but low update rates and high latency mean that it is impossible for applications in flight control. The proposed method can provide pose estimation with high sampling frequency and accuracy for the microFWRs platform with violent vibration during flight, which is essential for the agile maneuvering control of microFWRs. Based on bench tests, the proposed approach holds great promise for being generalized to agile MAVs with different scales and design principles. In the future, we will implement this method on different platforms to study its portability.

**Author Contributions:** Conceptualization, Z.T. and X.D.; methodology, Z.T., X.D. and F.F.; software, X.D.; validation, Z.W., F.L. and S.L.; formal analysis, X.D.; investigation, Z.W.; resources, D.L.; data curation, Z.W.; writing—original draft preparation, X.D.; writing—review and editing, Z.T. and F.F.; visualization, X.D.; supervision, D.L.; project administration, Z.T.; funding acquisition, Z.T. and D.L. All authors have read and agreed to the published version of the manuscript.

**Funding:** This research was funded by the National Natural Science Foundation of China, number 52102431, and the National Natural Science Foundation of China, number A020314.

**Institutional Review Board Statement:** Not applicable.

**Informed Consent Statement:** Not applicable.

**Data Availability Statement:** Not applicable.

**Acknowledgments:** The authors gratefully acknowledge the support of the National Natural Science Foundation of China. In addition, our thanks are given to the editors and reviewers for contributing to the final form of this research.

**Conflicts of Interest:** The authors declare no conflicts of interest.

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
