# Peer review of "Visual-Inertial Cross Fusion: A Fast and Accurate State Estimation Framework for Micro Flapping Wing Rotors"

_drones, doi:10.3390/drones6040090_

Round 1

Reviewer 1 Report

I reviewed the manuscript entitled  “Visual-Inertial Cross Fusion: A Fast and Accurate State Estimation Framework for Micro Flapping Wing Rotors” and found that the topic is interesting. The structure, the presentation, and the language are clear. However, some points need to be clarified and addressed:

  • A diagram that shows the methodology workflow is necessary
  • Have you considered an abrupt change in the dynamic model? If yes, how did you deal with it? The state covariance matrix P should be adapted if the dynamic model is unstable using filters such as strong tracking filter or captured filter.  Please elaborate
  • What does the observation vector (y) include? Please add the variables.
  • The length of navigation time is not clear. Is it 15 seconds? If yes, in Figure (13) you only presented 10 seconds out of the total length. Why?
  • The results were presented but without discussion.
  • The conclusions need additional information about your research findings

Reviewer 2 Report

With interest, I read the manuscript. It is appreciated that the manuscript is easy to follow and not too long. The message is clear and of interest to the community.

Some observations on the document are as follows:

In line 50: The word Optictrackk from the revised in reference 26 should be replaced by the word OptiTrack, as mentioned in line 137.

In line 115: the electronics on board the platform is indicated, could you indicate which microcontroller was used?

In line 185: It is mentioned that three communication protocols have been tested, could you specify which maximum cable length was used during the experimentation and indicate which communication protocol was selected for the rest of the experimentation?
